# Head Pose Estimation through Keypoints Matching between Reconstructed 3D Face Model and 2D Image

**DOI:** 10.3390/s21051841

**Published:** 2021-03-06

**Authors:** Leyuan Liu, Zeran Ke, Jiao Huo, Jingying Chen

**Affiliations:** 1National Engineering Research Center for E-Learning, Central China Normal University, Wuhan 430079, China; lyliu@ccnu.edu.cn (L.L.); kezr@mails.ccnu.edu.cn (Z.K.); huojiao@mails.ccnu.edu.cn (J.H.); 2National Engineering Laboratory for Educational Big Data, Central China Normal University, Wuhan 430079, China

**Keywords:** computer vision, head pose estimation, 3D face reconstruction, facial keypoints matching

## Abstract

Mainstream methods treat head pose estimation as a supervised classification/regression problem, whose performance heavily depends on the accuracy of ground-truth labels of training data. However, it is rather difficult to obtain accurate head pose labels in practice, due to the lack of effective equipment and reasonable approaches for head pose labeling. In this paper, we propose a method which does not need to be trained with head pose labels, but matches the keypoints between a reconstructed 3D face model and the 2D input image, for head pose estimation. The proposed head pose estimation method consists of two components: the 3D face reconstruction and the 3D–2D matching keypoints. At the 3D face reconstruction phase, a personalized 3D face model is reconstructed from the input head image using convolutional neural networks, which are jointly optimized by an asymmetric Euclidean loss and a keypoint loss. At the 3D–2D keypoints matching phase, an iterative optimization algorithm is proposed to match the keypoints between the reconstructed 3D face model and the 2D input image efficiently under the constraint of perspective transformation. The proposed method is extensively evaluated on five widely used head pose estimation datasets, including Pointing’04, BIWI, AFLW2000, Multi-PIE, and Pandora. The experimental results demonstrate that the proposed method achieves excellent cross-dataset performance and surpasses most of the existing state-of-the-art approaches, with average MAEs of $4.78^{\circ }$ on Pointing’04, $6.83^{\circ }$ on BIWI, $7.05^{\circ }$ on AFLW2000, $5.47^{\circ }$ on Multi-PIE, and $5.06^{\circ }$ on Pandora, although the model of the proposed method is not trained on any of these five datasets.

## 1. Introduction

Head pose plays a significant role in diverse applications such as human–computer interaction [1], driver monitoring [2], and analysis of students’ learning state [3], since it usually indicates the gaze direction and even the attention of a person. Moreover, head pose provides an important cue for many other face-related computer vision tasks, including facial feature points’ detection [4], multi-view face recognition [5], and facial expression analysis [6,7]. As a consequence, head-pose estimation has become a hot topic in the computer-vision community. Inaccurate estimations of head pose may indicate wrong gaze direction or attention, and thus result in poor user experience for almost all applications. For example, in the application of students’ learning-state analysis [3], inaccurate head-pose estimations often lead to incorrect learning states. Hence, from the classical AdaBoost-based [8] and random-forests-based [9,10] methods to the current deep neural networks [11,12,13,14], hundreds of methods have been proposed to pursuit more accurate head-pose estimations.

The most recent mainstream methods [9,10,14,15] treat head-pose estimation as a supervised classification/regression problem, and need to train models on large-scale datasets with ground-truth head-pose labels. As it is often assumed that a person’s head is a rigid object with three degrees of freedom, the label of head pose is characterized by the pitch, yaw and roll angles with respect to the image plane [16]. From the machine learning perspective, the task of head-pose estimation consists of learning a model that maps an input head image to head-pose angles based on image and ground-truth label pairs. It is well-known that the maximum achievable accuracy of a supervised model depends on the quality of the training data [17]. Like the other supervised machine learning tasks, the performance of supervised-learning-based head-pose estimation methods also heavily depends on the accuracy of the ground-truth labels in the training dataset [14,15].

However, it is difficult to obtain accurate head-pose labels in practice [14,15]. Since there is a lack of effective equipment and reasonable approaches for head-pose labeling, approximate approaches are usually adopted to label head-pose angles in the training dataset. Take the labeling approach of the widely used Pointing’04 head pose dataset [18] as an example: First, put 93 markers around a half-sphere of a chair in a room. Each marker corresponds to a head pose, characterized by a pitch angle and a yaw angle. Then, ask each subject to sit on the chair in the center of the half-sphere, and then to stare successively at each of the markers. Finally, take an image when the subject is staring at a certain marker, and use the pose that the marker stands for as the head-pose label. This head-pose labeling approach has at least two disadvantages: (1) Only a few coarse discrete head poses can be labeled. (2) The head-pose labels are pretty inaccurate, because the subject tends to squint at the markers. Multiple-camera array, automatic algorithms, or even subjective consciousness of data annotators are adopted to annotate head-pose labels in other datasets. Hence, besides the Pointing’04 dataset, inaccurate labels are common in other head-pose estimation datasets. Although many works [10,14] have reported promising head-pose estimation performance, it is worth noting that these methods are trained and evaluated on datasets with inaccurate or even wrong labels. The methods trained and tested on the same dataset with inaccurate labels tend to create an illusion that their models can accurately estimate head pose. Unfortunately, a cross-dataset testing can easily break the illusion. Therefore, head-pose estimation methods with robust cross-dataset performance still need to be explored.

To avoid suffering from inaccurate labels in training datasets, a head-pose estimation method that employs keypoint-matching between the input image and the corresponding reconstructed 3D face model is proposed in this paper. The key insight of the proposed method is that if a personalized 3D face model is reconstructed from the input head image, then the keypoints between the 3D face model and input image can be matched under the constraint of perspective transformation, and finally the head pose angles can be estimated by the perspective transformation parameters. Because the proposed method does not use any head-pose label during training, it avoids suffering from the inaccuracy of head-pose labels and exhibits outstanding cross-dataset ability. Experimental results show that the proposed method achieves excellent cross-dataset performance on a serial of the widely used benchmark datasets including Pointing’04 [18], BIWI [19], AFLW2000 [20], Multi-PIE [21], and Pandora [22], as illustrated in Figure 1.

The main contribution of this paper is three-fold:

(1) A head-pose estimation method with excellent cross-dataset performance is proposed. Unlike most supervised learning-based head-pose estimation methods that heavily depend on the accuracy of ground-truth labels, our method does not use any head-pose label during the training, and thus avoids suffering from the inaccuracy labels which are common in most head-pose datasets;

(2) Convolutional neural networks which are jointly optimized by an asymmetric Euclidean loss and keypoints loss is proposed to reconstruct the personalized 3D face model from a single input head image. The 3D face models reconstructed by our networks have accurately aligned 3D facial keypoints, which are beneficial for head-pose estimation;

(3) An iterative optimization algorithm, which consists of an R-step and a ST-step, is proposed to match the keypoints between the reconstructed 3D face model and the 2D input image under the constraint of weak perspective transformation. This iterative algorithm is not only efficient, but also effective in obtaining high-quality 3D-2D keypoint-matching results.

The rest of the paper is arranged as follows: the related work is introduced in Section 2, the details of the proposed head-pose estimation method are described in Section 3, the experiments are presented in Section 4, and the conclusions are given in Section 5.

## 2. Related Work

### 2.1. Supervised-Learning-Based Methods

Supervised-learning-based methods learn models that map input head images to head-pose angles based on ground-truth labels. Fanelli et al. [9] formulated the problem of head-pose estimation as a regression problem, and unutilized random regression forests for estimating head pose from depth data. Experimental results show that this approach can handle depth images with partial occlusions and facial expressions. Inspired by the work of Fanelli et al. [9], Liu et al. [10] proposed Dirichlet-tree distribution enhanced random forests to estimate head pose from RGB images captured in an unconstrained environment. This approach treats head-pose estimation as a supervised classification problem, and reaches an average accuracy rate of 76.2% with 25 head-pose classes on Pointing’04. Although random forests have shown effectiveness and robustness [21,22] in head-pose estimation, experimental results reported by these approaches illustrate that their accuracies are too low to meet the requirements of practical applications.

Recently, deep neural networks have shown performance improvement in head-pose estimation. Ahn et al. [13] proposed a multi-task network to detect multi-view faces and jointly estimate head pose. Xu et al. [14] introduced a regularized convolutional neural network architecture that is optimized by Kullback–Leibler divergence loss and Jeffreys divergence loss. This method achieves an accuracy of 85.77% on Pointing’04. Bao et al. [23] used a three-level network architecture for head-pose estimation. Change et al. [24] proposed the use of a simple convolutional neural network for head-pose estimation, and showed that simple convolutional neural networks can be trained to accurately and robustly regress head pose. Kumar et al. [25] presented an architecture called H-CNN which captures structured global and local features to jointly estimate head pose and facial keypoints. Ruiz et al. [26] presented a method to estimate head pose by training a multi-loss convolutional neural network on a large synthetically expanded dataset. Inspired by SSR-Net [27] which can cast the regression problem as a classification problem, Yang et al. [28] proposed a soft stage-wise regression scheme and applied it to estimate head pose. Recently, Han et al. [29] developed a compact CNN-based approach for head-pose estimation. In order to extract more representative features, an attention model that includes both the spatial and channel attention structures is embedded into the compact CNN. In [22], a framework called POSEidon is proposed to estimate head pose using multimodal data. The POSEidon first utilizes a face-from-depth model to reconstruct gray-level face images from the depth maps captured by a Kinect V2, and then fusing the feature output by three independent convolutional nets, which, respectively, extract visual features from depth maps, RGB images, and the reconstructed gray-level images, for regressing head poses. Although these deep-learning-based methods have reported promising head-pose estimation results, some of the authors [14,24] have noticed that the inaccuracy of head-pose labels in the training and testing datasets may cause biased evaluations.

To avoid suffering from the inaccurate labels in training datasets, many researchers proposed synthesizing head-pose images and ground-truth labels. Liu et al. [30] proposed a method to generate a head-pose dataset by rendering images from 37 3D head models. Sun et al. [31] buildt a large-scale head-pose dataset including more than 140,000 images with diverse and accurate head poses. Although large-scale datasets with accurate labels can be generated, the current synthesized images are not realistic enough to train robust head-pose estimation models. Xu et al. [14] and Geng et al. [15] proposed generating soft labels to improve the accuracy of ground-truth labels. However, they constructed soft labels using a Gaussian distribution function. In practice, the error of the labels in the training dataset does not usually obey the Gaussian distribution.

### 2.2. Model-Based Methods

Model-based methods utilize the geometric distortion of facial keypoints or 3D morphable face models to infer head-pose angles from VIS (RGB or gray-level) images. Feng et al. [32] proposed a method for joint 3D face reconstruction and dense alignment. In this method, a 2D representation called the UV position map is designed to record the 3D shape of a face in UV space, and a convolutional neural network is developed to regress the 3D face from a single 2D image. Gecer et al. [33] utilized generative adversarial networks to train a generator of facial texture in UV space, and then revisited the original 3D morphable model’s fitting approaches to find the optimal latent parameters that best reconstruct the 3D face model from the input 0image. Yu et al. [34] presented a framework which combines the strengths of a 3D morphable model fitted online with a prior-free reconstruction of a 3D full-head model, providing support for head-pose estimation. Although this method achieves a head-pose estimation accuracy of 94.7% on the BIWI dataset, which consists of RGB-D images captured by Kinect, its performance on RGB images is not evaluated. Recently, Andrea et al. [35] proposed exploiting a quad-tree based representation of facial features and estimate head pose by guiding the subdivision of the locations of a set of landmarks detected over the face image into smaller and smaller quadrants. In [36], a web-shaped model was proposed to associate each of them to a specific face sector over the detected landmarks. Although this method does not need to train on datasets with head-pose labels, it performs poorly under large poses. Zhu et al. [37] proposed an alignment framework termed 3D Dense Face Alignment (3DDFA), in which a dense 3D morphable model is fitted to the image via cascaded convolutional neural networks. This method achieves excellent head-pose estimation performance on RGB-image-based datasets like AFLW2000, but its cross-dataset performance still needs to be improved.

With the development of consumer-level depth-image sensors, many studies have tried to exploit 3D-face-model-based approaches on depth and 3D data. Martin et al. [38] proposed a real-time head-pose estimation approach on consumer depth cameras. This approach first creates a point-cloud-based 3D head model from the input depth image, and then registers the 3D head model with the iterative closest point (ICP) algorithm for head-pose estimation. Gregory et al. [39] proposed estimating head poses by registering a 3D morphable model (3DMM) to the input depth data through a combination of particle swarm optimization (PSO) and the ICP algorithm. Instead of creating a 3D face model for a subject at one stroke, this method dynamically adapts the weights of the 3DMM to fit the subject’s face on the fly. Unlike the 3D model registration methods, Papazov et al. [40] introduced a triangular surface patch (TSP) to represent the shape of the 3D face surface within a triangular area, and utilized the matched triangular surface patches in the training set to estimate the head pose. Although estimating the head poses on the depth image can avoid suffering from the cluttered background and illumination changes that are common in RGB images, depth image sensors are not available in most of the current applications.

## 3. Methodology

### 3.1. Overview

The proposed head pose estimation method, at its core, consists of two phases, i.e., 3D face reconstruction and 3D–2D keypoints matching, as illustrated in Figure 2. In the 3D face reconstruction phase, a personalized 3D face model is reconstructed from the input head image using convolutional neural networks which are jointly optimized by an asymmetric Euclidean loss and a keypoint loss. In the 3D–2D keypoints matching phase, an iterative optimization algorithm is proposed to match the keypoints between the reconstructed 3D face model and the 2D input image efficiently under the constraint of weak perspective transformation.

### 3.2. 3D Face Model Reconstruction

#### 3.2.1. Model Representation

In our method, a 3D face model is represented by a mesh with *N* vertices
(1)$$P={\left[p_{1},p_{2},\cdots ,p_{N}\right]}^{T}$$
where $p_{n}=\left(x_{n},y_{n},z_{n}\right)$ denotes the 3D location of a vertex. In practice, a 3D face model usually consists of hundreds of thousands of vertices. That means that P is high-dimensional, which makes direct reconstruction of a 3D face model an extremely difficult problem. In order to reduce the difficulty of the problem, Principal Component Analysis (PCA) is employed to encode the high-dimensional 3D face model on lower-dimensional subspaces
(2)$$P=\bar{P}+M_{s}\alpha_{s}$$
where $\bar{P}={\left[{\bar{p}}_{1},{\bar{p}}_{2},\cdots ,{\bar{p}}_{N}\right]}^{T}$ is the mean 3D face computed over a set of 3D face meshes, $M_{s}$ is the principal components matrix of the same set of 3D face meshes, $\alpha_{s}$ is the coefficient vector that characterizes the geometry of a specific 3D face. The dimensions of $\bar{P}$, M, and α are respectively 3N×1, $3N\times d_{s}$ and $d_{s}\times 1$, and $d_{s}<<3N$. In the publicly available Basel Face Model [41], $\bar{P}$ and $M_{s}$ have been computed over a dataset of aligned facial 3D face scans.

In practice, the facial keypoints of a subject can be changed drastically due to facial expressions. Hence, we add an additional blend item to represent facial expressions
(3)$$P=\bar{P}+M_{s}\alpha_{s}+M_{e}\alpha_{e}$$
where $M_{e}$ is the principle matrix extracted from the offsets between the expression meshes and the neutral meshes, and $\alpha_{e}$, which is $d_{e}$-dimensional, is the expression parameter of a given face image. In our method, we use the FaceWarehouse [42] dataset for computing $M_{e}$. At this point, to reconstruct a 3D face model from a 2D RGB image, we only need to estimate the low-dimensional geometric parameter $\alpha_{s}$ and expression parameter $\alpha_{e}$.

#### 3.2.2. Network Structure

As illustrated in Figure 3, two convolutional neural networks are utilized to regress the geometric parameter $\alpha_{s}$ and the expression parameter $\alpha_{e}$, respectively, from the input image. We adopt ResNet-101 and ResNet-18 [43] as the backbone networks, due to their promising performance on many face-related computer-vision tasks. The input of the networks is a 2D RGB head image whose size is 224×224, and the outputs of the fully connected layers of the two ResNets are, respectively, modified to $d_{s}$-dimensional and $d_{e}$-dimensional for regressing the geometric parameter $\alpha_{s}$ and the expression parameter $\alpha_{e}$. The regressed geometric parameter $\alpha_{s}$ and expression parameters $\alpha_{e}$ are then passed into a PCA reconstruction module for reconstructing the 3D face model.

The backbone ResNet-101 and ResNet-18 are first pre-trained on the CASIA-WebFace dataset [44], and then fine-tuned on the 3DFaceNet dataset [45]. In the 3DFaceNet dataset, the geometric parameters under neutral expression are extracted based on the BaselFace model, and the offsets between the expression meshes and the neutral meshes are also computed based on the FaceWarehouse model. Hence, the ResNet-101 and ResNet-18 can be trained on 3DFaceNet for regressing the geometric parameter $\alpha_{s}$ and the expression parameter $\alpha_{e}$.

#### 3.2.3. Loss Functions

Networks trained with direct MSE (Mean Square Error) loss or Euclidean loss on the model parameter α tends to generate 3D faces that are similar to the mean face $\bar{P}$. To counter this bias towards the mean face, the asymmetric Euclidean loss [46] is utilized in our networks
(4)$$\;L_{E}\left(\hat{\alpha },\alpha \right)=\lambda_{1}{∥\alpha^{+}-\alpha_{\max}∥}^{2}+\lambda_{2}{∥{\hat{\alpha }}^{+}-\alpha_{\max}∥}_{2}^{2}$$
where $\alpha \in \{\alpha_{s},\alpha_{e}\}$, $\hat{\alpha }\in \left\{{\hat{\alpha }}_{s},{\hat{\alpha }}_{e}\right\}$ is the estimated model parameters output by the two networks, $\alpha^{+}=\mathrm{sign}\left(\alpha \right)\cdot \alpha$, ${\hat{\alpha }}^{+}=\mathrm{sign}\left(\hat{\alpha }\right)\cdot \hat{\alpha }$, and $\alpha_{\max}=\max\left({\hat{\alpha }}^{+},\alpha^{+}\right)$. Since the asymmetric Euclidean loss encourages the network to favor estimates further away from the mean face, the networks trained with the asymmetric Euclidean loss can generate more diverse 3D faces. However, we found that networks trained with the asymmetric Euclidean loss still struggle to output 3D faces with accurate facial components. The performance of our head-pose estimation method will be affected by the inaccurate estimated facial components, since our method is based on facial keypoint-matching between a 2D face image and the corresponding reconstructed 3D face model.

In order to make our networks generate 3D face models with accurate facial components, we propose adding additional Facial Feature Points (FFP) loss to align the facial components between the estimated 3D face and the corresponding ground-truth. The additional FFP loss is based on the facial feature points (i.e., the 3D vertices around facial components such as the eyes, mouth, and nose). For the sake of representation convenience, we rewrite Equation (Equation 3) as
(5)$$\left[\begin{array}{c} p_{1}\\ p_{2}\\ \vdots \\ p_{N} \end{array}\right]=\left[\begin{array}{c} {\bar{p}}_{1}\\ {\bar{p}}_{2}\\ \vdots \\ {\bar{p}}_{N} \end{array}\right]+\left[\begin{array}{c} M_{s}^{1}\\ M_{s}^{2}\\ \vdots \\ M_{s}^{N} \end{array}\right]\alpha_{s}+\left[\begin{array}{c} M_{e}^{1}\\ M_{e}^{2}\\ \vdots \\ M_{e}^{N} \end{array}\right]\alpha_{e}$$
then, the $n^{th}$ vertex of a 3D face mesh can be denoted as
(6)$$p_{n}={\bar{p}}_{n}+M_{s}^{n}\alpha_{s}+M_{e}^{n}\alpha_{e}$$

Denote all the facial keypoints selected for computing the FFP loss as $\tilde{P}=\left\{{\tilde{p}}_{k}\right\}$. The FFP loss computed by a single facial feature point ${\tilde{p}}_{k}$ is defined by the distance between its estimation and ground-truth
(7)$$\begin{array}{cc} L_{{\tilde{p}}_{k}}\left(\hat{\alpha },\alpha \right) & ={∥{\tilde{p}}_{k}-{\hat{p}}_{k}∥}_{2}^{2}\\ & ={∥M_{s}^{k}\left({\hat{\alpha }}_{s}-\alpha_{s}\right)+M_{e}^{k}\left({\hat{\alpha }}_{e}-\alpha_{e}\right)∥}_{2}^{2} \end{array}$$

Then, the FFP loss computed by all the selected facial keypoints is
(8)$$L_{\tilde{P}}\left(\hat{\alpha },\alpha \right)=\frac{1}{|\tilde{P}|}\sum_{p_{n}\in \tilde{P}}^{}{∥M_{s}^{n}\left({\hat{\alpha }}_{s}-\;\alpha_{s}\right)+M_{e}^{n}\left({\hat{\alpha }}_{e}-\alpha_{e}\right)∥}_{2}^{2}$$

The final loss function is a weighted sum of the asymmetric Euclidean loss and the FFP loss
(9)$$L\left(\hat{\alpha },\alpha \right)=\omega L_{E}\left(\hat{\alpha },\alpha \right))+\left(1-\omega \right)L_{\tilde{P}}\left(\hat{\alpha },\alpha \right)$$
where ω is the weight parameter.

### 3.3. Head-Pose Estimation through 3D-2D Keypoints Matching

#### 3.3.1. Weak Perspective Transformation

Given a reconstructed 3D face model, the weak perspective transformation is employed to project vertices (3D points) on the 3D face model onto the 2D image plane
(10)q=sΠR(θ)p+t
where $p={\left(p_{x},p_{y},p_{z}\right)}^{T}$ and $q={\left(q_{x},q_{y}\right)}^{T}$ are, respectively, the locations of a 3D point on the reconstructed face model and the projected 2D point on the image plane, $\Pi =\left(\begin{array}{ccc} 1 & 0 & 0\\ 0 & 1 & 0 \end{array}\right)$ is the perspective matrix, *s* is the scale factor, $t={\left[t_{x},t_{y}\right]}^{T}$ is the translation parameter, and R(θ) is the rotation matrix characterized by the underlying head pose θ. As it is often assumed that a person’s head is a rigid object with three degrees of freedom, a head pose (θ) is usually denoted by a pitch angle ($\theta_{p}$), a yaw angle ($\theta_{y}$) and, a roll angle ($\theta_{r}$), with respect to the image plane, i.e., $\theta ={\left[\theta_{p},\theta_{y},\theta_{r}\right]}^{T}$. Under the weak perspective transformation, R(θ) can be decomposed as $R\left(\theta \right)=R_{x}\left(\theta_{r}\right)R_{y}\left(\theta_{p}\right)R_{z}\left(\theta_{y}\right)$, where
(11)$$\begin{array}{c} R_{x}\left(\theta_{r}\right)=\left(\begin{array}{ccc} 1 & 0 & 0\\ 0 & cos\left(\theta_{r}\right) & sin\left(\theta_{r}\right)\\ 0 & -sin\left(\theta_{r}\right) & cos\left(\theta_{r}\right) \end{array}\right)\\ R_{y}\left(\theta_{p}\right)=\left(\begin{array}{ccc} cos\left(\theta_{p}\right) & 0 & -sin\left(\theta_{p}\right)\\ 0 & 1 & 0\\ sin\left(\theta_{p}\right) & 0 & cos\left(\theta_{p}\right) \end{array}\right)\\ R_{z}\left(\theta_{y}\right)=\left(\begin{array}{ccc} cos\left(\theta_{y}\right) & sin\left(\theta_{y}\right) & 0\\ -sin\left(\theta_{y}\right) & cos\left(\theta_{y}\right) & 0\\ 0 & 0 & 1 \end{array}\right) \end{array}$$

#### 3.3.2. 3D–2D Keypoints Matching

After projecting the 3D keypoints on the reconstructed face model onto the image plane, the projected keypoints and the corresponding keypoints detected from the input image are matched to infer the underlying head pose (s shown in Figure 4a). The total Euclidean distance of all pairs of keypoints are adopted to evaluate the matching result. To this end, 3D–2D keypoint-matching is formulated as an optimization problem that minimizes the following energy
(12)$$E\left(s,t,\theta \right)=\sum_{i=1}^{K}{∥q_{i}-{\tilde{q}}_{i}∥}_{2}^{2}=\sum_{i=1}^{K}{∥s\Pi R\left(\theta \right)p_{i}+t-{\tilde{q}}_{i}∥}_{2}^{2}$$
where *K* is the number of selected keypoints, $q_{i}$ is the projected 2D keypoint, and ${\tilde{q}}_{i}$ is the corresponding keypoint detected by a certain facial keypoint detector. Hence, in our method, head-pose estimation is transformed to solve the following optimization problem
(13)$$arg\;\underset{s,t,\theta }{min}E\left(s,t,\theta \right)$$

Obviously, this is an unconstrained nonlinear optimization problem with six parameters: $\left\{s,\left(t_{x},t_{y}\right),\left(\theta_{p},\theta_{y},\theta_{r}\right)\right\}$.

To simplify the above optimization problem, the six parameters are divided into two groups (i.e., $\left\{s,t_{x},t_{y}\right\}$ and $\left\{\theta_{p},\theta_{y},\theta_{r}\right\}$) and optimized in an iterative manner. As illustrated in Figure 4b, we propose an ST-step and a R-step to alternatively optimize the two groups of parameters until the energy is lower than a given threshold (ϵ):(14)$$E(\tilde{s},\tilde{t},\tilde{\theta })<\epsilon$$
where $\tilde{s}$ and $\tilde{t}={\left[{\tilde{t}}_{x},{\tilde{t}}_{y}\right]}^{T}$ are, respectively, the optimal s and t output by the ST-step, and $\tilde{\theta }={\left[\tilde{\theta_{p}},\tilde{\theta_{y}},\tilde{\theta_{r}}\right]}^{T}$ are the optimal $\theta_{p},\theta_{y}$, and $\theta_{r}$ output by the R-step.

**ST-step.** In this step, we fix $\left\{\theta_{p},\theta_{y},\theta_{r}\right\}$ to $\left\{\tilde{\theta_{p}},\tilde{\theta_{y}},\tilde{\theta_{r}}\right\}$ and optimize $\left\{s,t_{x},t_{y}\right\}$. The optimization problem of ST-step is formulated as
(15)$$arg\;\underset{s,t}{min}E_{ST}\left(s,t|\tilde{\theta }\right)$$
and the energy function is
(16)$$E_{ST}\left(s,t|\tilde{\theta }\right)=\sum_{i=1}^{K}{∥s\Gamma_{i}\left(\tilde{\theta }\right)+t-{\tilde{q}}_{i}∥}_{2}^{2}$$
where $\Gamma_{i}\left(\tilde{\theta }\right)=\Pi R\left(\tilde{\theta }\right)p_{i}$. Obviously, this is a linear least-squares problem. We set $\frac{\partial E_{ST}\left(s,t|\tilde{\theta }\right)}{\partial s}=0$, $\frac{\partial E_{ST}\left(s,t|\tilde{\theta }\right)}{\partial t_{x}}=0$, and $\frac{\partial E_{ST}\left(s,t|\tilde{\theta }\right)}{\partial t_{y}}=0$ which give the following equations
(17)$$\left\{\begin{array}{c} \sum_{i=1}^{K}s\Gamma_{i}^{T}\Gamma_{i}=\sum_{i=1}^{K}\Gamma_{i}^{T}\left({\tilde{q}}_{i}-t\right)\\ \;\sum_{i=1}^{K}s\left(\Lambda_{0}^{T}\Gamma_{i}\;+\Gamma_{i}^{T}\Lambda_{0}\right)\;=\;\sum_{i=1}^{K}\left(\Lambda_{0}^{T}\left({\tilde{q}}_{i}-\;t\right)+{\left({\tilde{q}}_{i}-t\right)}^{T}\Lambda_{0}\right)\\ \;\sum_{i=1}^{K}s\left(\Lambda_{1}^{T}\Gamma_{i}\;+\Gamma_{i}^{T}\Lambda_{1}\right)\;=\;\sum_{i=1}^{K}\left(\Lambda_{1}^{T}\left({\tilde{q}}_{i}-\;t\right)+{\left({\tilde{q}}_{i}-t\right)}^{T}\Lambda_{1}\right) \end{array}\right.$$
where $\Lambda_{0}={\left[1,0\right]}^{T}$, and $\Lambda_{1}={\left[0,1\right]}^{T}$. After solving the above linear ternary equations, the optimal $\left\{\tilde{s},{\tilde{t}}_{x},{\tilde{t}}_{y}\right\}$ are obtained.

**R-step.** In this step, we fix $\left\{s,t_{x},t_{y}\right\}$ to $\left\{\tilde{s},{\tilde{t}}_{x},{\tilde{t}}_{y}\right\}$ and optimize $\left\{\theta_{p},\theta_{y},\theta_{r}\right\}$. The optimization problem of R-step is formulated as
(18)$$arg\;\underset{\theta }{min}E_{R}\left(\theta |\tilde{s},\tilde{t}\right)$$
and the energy function is
(19)$$E_{R}\left(\theta |\tilde{s},\tilde{t}\right)=\sum_{i=1}^{K}{∥\tilde{s}\Gamma_{i}\left(\theta \right)+\tilde{t}-{\tilde{q}}_{i}∥}_{2}^{2}$$
where $\Gamma_{i}\left(\theta \right)=\Pi R\left(\theta \right)p_{i}$. This is a non-linear least-squares problem because of R(θ). To solve it, we first compute the Jacobian matrix
(20)$$\left[\begin{array}{ccccc} \frac{\partial \Gamma_{1}\left(\theta \right)}{\partial \theta_{p}} & \cdots & \frac{\partial \Gamma_{i}\left(\theta \right)}{\partial \theta_{p}} & \cdots & \frac{\partial \Gamma_{K}\left(\theta \right)}{\partial \theta_{p}}\\ \frac{\partial \Gamma_{1}\left(\theta \right)}{\partial \theta_{y}} & \cdots & \frac{\partial \Gamma_{i}\left(\theta \right)}{\partial \theta_{y}} & \cdots & \frac{\partial \Gamma_{K}\left(\theta \right)}{\partial \theta_{y}}\\ \frac{\partial \Gamma_{1}\left(\theta \right)}{\partial \theta r} & \cdots & \frac{\partial \Gamma_{i}\left(\theta \right)}{\partial \theta_{r}} & \cdots & \frac{\partial \Gamma_{K}\left(\theta \right)}{\partial \theta_{r}} \end{array}\right]$$
where
(21)$$\left\{\begin{array}{c} \frac{\partial \Gamma_{i}\left(\theta \right)}{\partial \theta_{p}}\;=\;\left(\begin{array}{ccc} -s_{p}c_{y} & -s_{p}s_{r} & -c_{p}\\ c_{p}c_{y}s_{r} & c_{p}s_{y}s_{r} & -s_{p}s_{r} \end{array}\right)p_{i}\\ \frac{\partial \Gamma_{i}\left(\theta \right)}{\partial \theta_{y}}\;=\;\left(\begin{array}{ccc} \;-c_{p}s_{y} & 0 & 0\\ \;-c_{y}c_{r}-s_{p}s_{y}s_{r} & -s_{y}c_{r}+s_{p}c_{y}s_{r} & 0 \end{array}\right)p_{i}\\ \frac{\partial \Gamma_{i}\left(\theta \right)}{\partial \theta_{y}}\;=\;\left(\begin{array}{ccc} \;0 & \;c_{p}c_{r} & \;0\\ \;s_{y}s_{r}+s_{p}c_{y}c_{r} & \;-c_{y}s_{r}+s_{p}s_{y}c_{r} & \;c_{p}c_{r} \end{array}\right)p_{i} \end{array}\right.$$
in which $c_{p}=cos\left(\theta_{p}\right)$, $c_{y}=cos\left(\theta_{y}\right)$, $c_{r}=cos\left(\theta_{r}\right)$, $s_{p}=sin\left(\theta_{p}\right)$, $s_{y}=sin\left(\theta_{y}\right)$, and $s_{r}=sin\left(\theta_{r}\right)$. With the Jacobian matrix, the optimal $\left\{\tilde{\theta_{p}},\tilde{\theta_{y}},\tilde{\theta_{r}}\right\}$ are then calculated using the *Levenberg–Marquardt* algorithm [47].

## 4. Experimental Results

### 4.1. Implementation Details

At the 3D face-reconstruction phase, the backbone networks employed in our method are first pre-trained on the CASIA-WebFace dataset [44], and then fine-tuned on the 3DFaceNet dataset [45]. The number of vertices on a 3D face mesh is 46,990, and the dimensions of the model parameters, i.e., the $\alpha_{s}$ and $\alpha_{e}$, are, respectively, set as 99 and 29. The batch size is fixed to 128 in the whole training phase. The initial learning rate is set to 0.05, and the iteration is set to 60 for each epoch. After 40 epochs, the learning rate is decayed to 0.005. Stochastic Gradient Descent (SGD) is adopted as the optimizer, and its initial momentum and weight-decay parameter are, respectively, set as to 0.9 and 0.0005. The backbone ResNet-101 is first trained independently for 200 epochs, and then the backbone ResNet-18 and ResNet-101 are jointly trained for another 100 epochs. At the 3D–2D keypoint-matching phase, the FAN [48], a CNN-based facial keypoints detector, is adopted to jointly detect faces and the 2D facial keypoints from the input images. The information of each 3D keypoint is obtained from the reconstructed 3D face model data according to the index of the keypoint.

### 4.2. Datasets and Performance Metric

The proposed head-pose estimation method is extensively evaluated on four public datasets, including Pointing’04 [18], BIWI [19], AFLW2000 [20], Multi-PIE [21], and Pandora [22]. The Pointing’04 dataset [18] contains 2790 face images of 15 subjects. In the original Pointing’04, only discrete yaw ($0^{\circ }$, $\pm 15^{\circ }$, $\pm 30^{\circ }$, $\pm 45^{\circ }$, $\pm 60^{\circ }$, $\pm 75^{\circ }$, $+90^{\circ }$) and pitch ($0^{\circ }$, $\pm 15^{\circ }$, $\pm 30^{\circ }$, $\pm 60^{\circ }$, $\pm 90^{\circ }$) angles are annotated as head-pose labels. As mentioned before, the head-pose labels in this dataset are annotated by asking subjects to stare successively at 93 markers, and therefore the precision of the original labels is low. The BIWI head-pose dataset [19] contains over 15,000 images of 20 subjects captured by a Kinect. Although an RGB image and a corresponding depth are provided for each frame, only the RGB images are used in our experiments. In this dataset, the head-pose range covers about ±75 degrees yaw and ±60 degrees pitch. The head-pose labels of the RGB images are automatically annotated by a head-pose estimation algorithm, [49], which works on depth images. The AFLW2000 dataset [20] contains the first 2000 identities of the AFLW dataset [50], which provides the large-scale collection of face images. As gathered from Flickr, the faces in AFLW have large pose variations, with various illumination conditions and expressions. The head-pose labels in AFLW2000 are labeled using a 3D-model-fitting approach [20]. The multi-PIE dataset [21] contains 755,370 images of 337 subjects, which are divided into four recording sessions. In our experiments, the 32,682 multi-view images in session0 are used for testing. Since the images in Multi-PIE are captured by 13 cameras, located at head height and spaced at $15^{\circ }$ intervals, 13 discrete yaw angles ($0^{\circ }$, $\pm 15^{\circ }$, $\pm 30^{\circ }$, $\pm 45^{\circ }$, $\pm 60^{\circ }$, $\pm 75^{\circ }$, $\pm 90^{\circ }$) are annotated as head-pose labels. The Pandora [22] is a dataset specifically created for head-center localization, head-pose and shoulder-pose estimation in a simulated automotive context. The Pandora contains 110 annotated sequences of more than 250,000 full-resolution RGB (1920 × 1080 pixels), and the corresponding depth images (512 × 424) captured by a Kinect V2. Although most of the existing methods fuse both the RGB and depth images to boost the performance of head-pose estimation on this dataset, we only employ the RGB images to evaluate our method in all experiments. In order to evaluate cross-dataset performance, our method is not trained on any of these five datasets.

**Ground-Truth Label Relabeling.** For fair and unprejudiced performance evaluation, the head-pose labels in the testing datasets are relabeled independently by three data annotators using a semi-automatic head-pose labeling tool provided by us. The pipeline of data relabeling is illustrated in Figure 5. First, for each input image, a 3D face model is reconstructed automatically by the method described in Section 3.2. Meanwhile, the 68 keypoints on the reconstructed 3D face model are highlighted. Then, the 68 corresponding keypoints in the input image are also marked using FAN [48]. If the keypoints marked by FAN are not accurate, the data annotator can edit them manually. Finally, the data annotator manipulates a keyboard to translate, scale, and rotate the 3D face model until the corresponding keypoints on the 3D face model and the input image are almost overlapped. As the labeling tool records the rotation angles of the 3D face model, the head-pose angles of the input image can be labeled as long as the data annotator stops his/her manipulation. The relabeled and original head-pose labels of some images which are randomly selected form the Pointing’04 dataset are shown in Figure 6. As can be seen, the head-pose angles relabeled using our labeling tool are much more accurate than the original ones. This tool is also suitable for labeling off-the-shelf head images. The relabeled head-pose labels are shared with the head-pose estimation community at https://github.com/Autccnu/headPoseLabels (accessed on 5 March 2021).

**Performance Metric.** The Mean Absolute Error (MAE) is adopted as the metric for evaluating the head-pose estimation performance achieved by different methods. The MAE for one head-pose angle (yaw, pitch, or roll) is defined as follows
(22)$$MAE=\frac{1}{N}\sum_{i=1}^{N}\left(\left|\hat{\theta }{}_{ik}-\theta_{ik}\right|\right)$$
and the average MAE for two or three head-pose angles is defined as
(23)$$\bar{MAE}=\frac{1}{KN}\sum_{k=1}^{K}\sum_{i=1}^{N}\left(\left|\hat{\theta }{}_{ik}-\theta_{ik}\right|\right)$$
where *N* is the number of samples in the testing dataset, *K* denotes the number of head-pose angles adopted for performance evaluation, $\theta_{ik}$ and ${\hat{\theta }}_{ik}$ are, respectively, the ground-truth head-pose angles and the estimations.

### 4.3. Performance Analysis of the Proposed Method

As described in Section 3, the proposed method consists of two components, i.e., 3D face reconstruction and 3D-2D keypoints matching. To evaluate their respective effectiveness, we conducted two experiments in this subsection. In the first experiment, the head-pose estimation performance of the proposed method with different loss functions in the 3D face-reconstruction component is evaluated. In the second experiment, the performance of the proposed method using different facial keypoints detection methods is tested.

**Performance with Different Loss Functions.** In the experiment setup, the FFP loss described in Section 3.2.3 is first added to, and then removed from, the 3D face reconstruction component, and FAN [48] with 30 keypoints is fixed for detecting facial keypoints from the input image. Table 1 shows the performance of the proposed method with different loss functions for 3D face reconstruction. Obviously, on all four testing datasets, the proposed method with the FFP loss performs better than the method without the FFP loss. For better understanding of the results, some reconstructed 3D faces, as well as the estimated head-pose angles, are illustrated in Figure 7. It can bee seen that some of the facial components are distorted when the 3D faces are reconstructed without the FFP loss, which consequently reduces the accurate of head-pose estimation.

**Performance with Different Keypoints Detectors.** In the experiment setup, three methods, including Dlib [51], Openface [52], and FAN [48], are adopted for detecting facial keypoints from the input image, and the loss functions in Equation(Equation 9) are fixed for 3D face reconstruction. Figure 8 illustrates the performance of the proposed method with different facial-keypoint-detection methods under different keypoint configurations on the super-datasets, which consist of all four testing datasets. Overall, the proposed method with FAN outperforms the proposed method with the other two keypoint detectors. No matter which of the three keypoint detectors is used, the proposed method achieves the lowest average MAEs under the 30-keypoints configuration. Using the 30-keypoints configuration, the average MAEs produced by the proposed method with different facial keypoint detectors range from 5.98 to 7.59, which indicates that the proposed method is not very sensitive to different facial keypoint detection methods under this configuration. The best performance is achieved by the proposed method with the FAN detector under the 30-keypoints configuration. Hence, an FAN with 30 keypoints is fixed for facial keypoints’ detection in the following experiments.

**Error Analysis.** The MAE distribution of different head-pose angles is illustrated in Figure 9. It can be seen that large errors occur when the input heads are placed under large yaw angles. Figure 10 shows the reconstructed 3D face models and the facial keypoints detected by three detectors under various head-pose angles. Obviously, the reconstructed 3D face model is robust under variations in head-pose angle, while the keypoint detection methods perform poorly under large head-pose angles. It can be speculated that the large head-pose estimation errors produced by the proposed method are mainly caused by the inaccuracy of facial keypoint detection. Hence, we predict that the performance of our head-pose estimation method can be further improved with progress in facial keypoint detection methods.

**Results on Occluded Images.** In order to evaluate the performance of the proposed methods on occluded images, we add an artificial occluded block with a random position on each of the testing images in the five testing datasets. The size of the occluded block on each image ranges from 1/8 to 1/4 of the face region. Some results on the occluded testing images produced by the proposed head-pose estimation method are shown in Figure 11. The quantitative results are also shown in Table 2. The MAEs only increase by 1.31${}^{\circ }$ on Pointing’04, 2.02${}^{\circ }$ on BIWI, 1.99${}^{\circ }$ on AFLW2000, 2.01${}^{\circ }$ on Multi-PIE, and 1.13${}^{\circ }$ on Pandora. These results confirm the reliability of our method in the occlusion case.

**Computational Load.** The proposed head-pose estimation method is evaluated on a computer with an NVIDIA GeForce 1080Ti GPU and a Core i5 CPU. The computational load for the modules and the whole system are illustrated in Table 3. Obviously, the 3D face reconstruction module consumes almost 95% of the computing resources. However, for the applications that use video as an input, the 3D face reconstruction module only needs to be executed once for each subject. In such a scenario, our method can process at more than 10 FPS, even on low-cost computers.

### 4.4. Comparisons with Other Methods

**Comparisons on the Pointing’04 Dataset.** The proposed head-pose estimation method is compared with MGD [53], kCovGa [54], CovGA [54], CNN [55], fuzzy [56], MSHF [57], SAE-XGB [58], Hopenet [26], FSA-Net [27], hGLLiM [59], 3DDFA [37] and 4C_4S_var4 [36]. Among them, MGD, kCovGa, CovGA, CNN, fuzzy, MSHF, hGLLiM, and SAE-XGB have been trained on the Pointing’04 dataset following a five-fold cross-validation protocol; Hopenet and FSA-Net have been trained on another dataset called 300W-LP, while 3DDFA, 4C_4S_var4 and our method have not been trained with any head-pose label. Table 4 shows the performance of different methods. Obviously, the methods that were trained on the testing dataset achieved much better results than the methods that were not trained on the dataset, excepting 3DDFA and our method. Since Hopenet and FSA-Net are supervised-learning-based methods that are sensitive to the label biases of different datasets, these two methods perform poorly with MAEs of 23.10 and 21.96. Our method achieves MAEs of 4.30 and 5.27 on yaw angle and pitch angle, and outperforms all the other compared methods.

**Comparisons on the BIWI Dataset.** The proposed method is also compared with state-of-the-art methods, including CNN-syn [30], DNN [60], regression [61], Two-Stage [62], KEPLER [25], QuatNet [63], Dlib [51], FAN [48], 3DDFA [37], QT_PYR [35], 4C_4S_var4 [36], Haar-Like(LBP) [64] and HAFA [65] on the BIWI dataset. We divided these 14 methods into two groups. The first group included the supervised-learning-based methods, which were trained on the BIWI dataset following a five-fold cross-validation protocol, and the second group included the model-based methods, which were not trained on the BIWI dataset. Table 5 shows the performance of different methods. It can be seen that our method achieved the best average MAE of 6.83 on the BIWI dataset, except for the recent advanced method proposed in [36]. Although FAN [48] and our method employ the same 2D facial keypoint detector, our method achieves a much lower MAE of 5.81 on yaw angle. Despite 3DDFA reporting an outstanding performance on Pointing’04 (See Table 4), it performed poorly on BIWI, with an average MAE of 24.20, while our methods achieved an excellent performance on both Pointing’04 and BIWI.

**Comparisons with the Pandora Dataset.** The proposed method is compared with the SingleCNN [22], DoubleCNN [22], POSEidon [22], Hopenet [26], FSA-Net [27], and 3DDFA [37] on the Pandora dataset. Since Pandora is a multi-modal dataset, the DoubleCNN and POSEidon employ multiple modals including the depth map, motion RGB images, and FfD gray-level image, for head-pose estimation on this dataset. As illustrated in Table 6, multi-modal data help to improve the performance of head-pose estimation. However, our method outperformed the multi-modal-based methods, although it only uses a single RGB image as input.

### 4.5. Cross-Dataset Experiments

In this subsection, we present cross-dataset experiments conducted on five datasets including Pointing’04, BIWI, AFLW2000, Multi-PIE, and Pandora. Three state-of-the-art methods with released training and testing codes, including Hopenet [26], FSA-Net [27], and 3DDFA [37], were chosen for comparison. In the cross-dataset experiment setup, the FAS-Net and Hopenet were trained on another dataset with head-pose labels called 300W-LP [20]; the 3DDFA and our method were trained on a 300W-LP and CASIA WebFace dataset, respectively [44] and did not use head pose labels during training. In other words, none of these four methods were trained on any of the textcolorredfive testing datasets.

The cross-dataset experimental results are illustrated in Figure 12. The average MAEs of yaw and pitch angles achieved by each method are marked on the top of the bars. The mean values and standard deviations of MAEs on the super-dataset, which consists of all four testing datasets, are also calculated and shown in Figure 12. Although the FSA-Net outperformed our method by MAEs of 2.26 and 0.18 on BIWI and AFLW2000, respectively, it performed rather poorly on Pointing’04 and Multi-PIE. The Hopenet surpassed our method by a narrow margin on BIWI and AFLW, but its MAE on Pointing’04 is as high as 23.10. The 3DDFA only exceeded our method by an MAE of 1.45 on AFLW2000, while it reported a bad performance on BIWI and MultiPIE, with MAES of 24.20 and 10.81. It is obvious that our method performs steadily on all five testing datasets and achieved the smallest mean and standard variance of MAEs. The experimental results indicate that our method achieves a more robust cross-dataset performance than the other three competitive methods.

## 5. Conclusions

In this paper, a head-pose estimation method that uses keypoint-matching between the reconstructed 3D face model and 2D input image has been proposed. First, a personalized 3D face model is reconstructed from the input image using convolutional neural networks, which are jointly optimized by an asymmetric Euclidean loss and a keypoint loss. Then, keypoints between the 3D face model and input image are matched under the constraint of the weak perspective transformation by a effective iterative optimization algorithm. Finally, the head pose is estimated by the perspective transformation parameters. To evaluate the performance of the proposed and other current state-of-the-art methods, extensive experiments have been conducted on five widely used datasets, including Pointing’04, BIWI, AFLW2000, Multi-PIE, and Pandora. The experimental results illustrate that the proposed method achieves excellent cross-dataset performance on a set of the widely used benchmark datasets, while the other compared methods, on average, show bias on different datasets.Because the proposed method does not use any ground-truth head-pose label during training, it does not suffer from the inaccuracy of head-pose labels which exist in most publicly available training datasets. Hence, we suggest exploiting model-based methods for head-pose estimation if the labels in the training dataset are not accurate. We also suggest that cross-dataset experiments should be conducted to evaluate the performance of head-pose estimation methods.

## Figures and Tables

**Figure 1 sensors-21-01841-f001:**
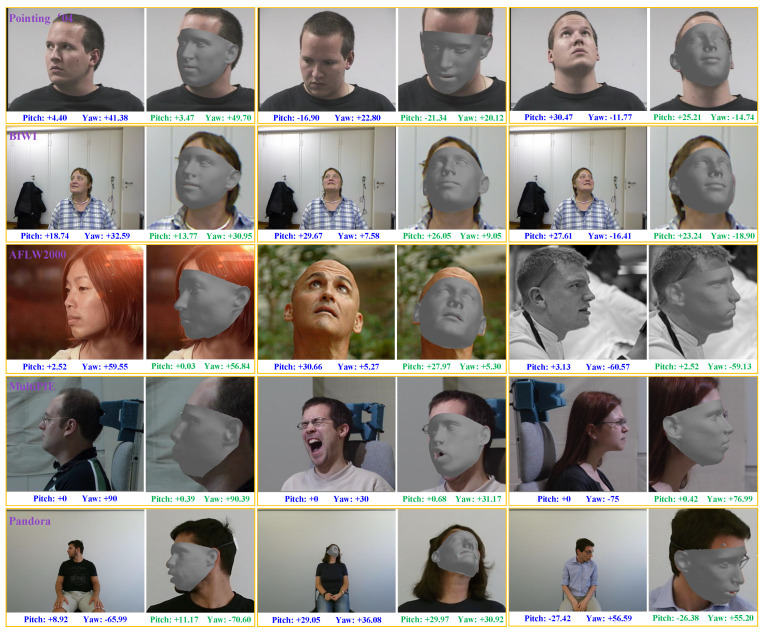
Head-pose estimation results produced by the propose method on the Pointing’04 (the 1st row), BIWI(the 2nd row), AFLW2000 (the 3rd row), Multi-PIE (the 4th row), and Pandora (the 5th row) datasets. The proposed method is not trained on any of these five datasets. The ground-truth labels and the estimated head pose angles are, respectively, marked in blue and green under each image.

**Figure 2 sensors-21-01841-f002:**
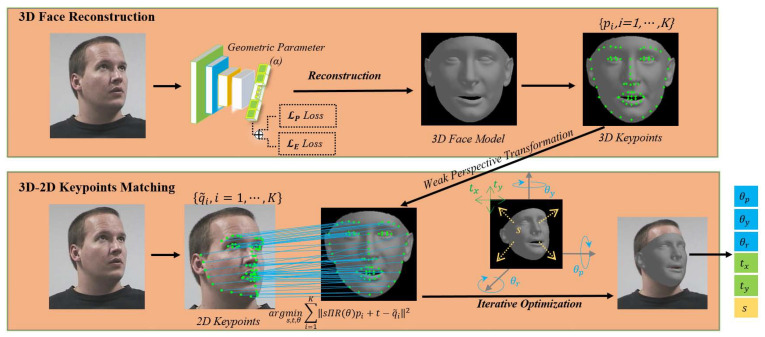
Framework of the proposed head-pose estimation method.The proposed method consists of two phases, i.e., 3D face reconstruction and 3D–2D keypoints matching. At the 3D face reconstruction phase, a personalized 3D face model is reconstructed from the input head image using a convolutional neural network, which is jointly optimized by an asymmetric Euclidean loss and a keypoint loss. At the 3D–2D keypoints matching phase, an iterative optimization algorithm is proposed to match the keypoints between the 2D input image and the reconstructed 3D face model under the constraint of weak perspective transformation.

**Figure 3 sensors-21-01841-f003:**
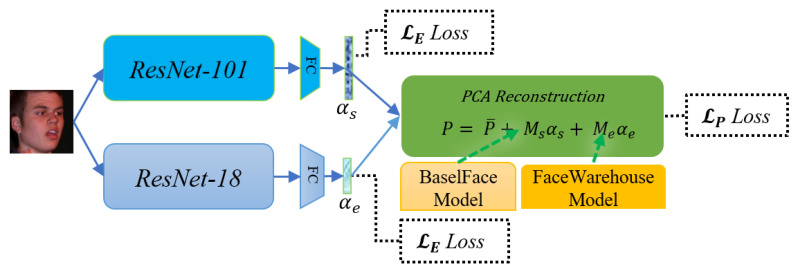
Structure of the deep neural networks employed for 3D face reconstruction.

**Figure 4 sensors-21-01841-f004:**
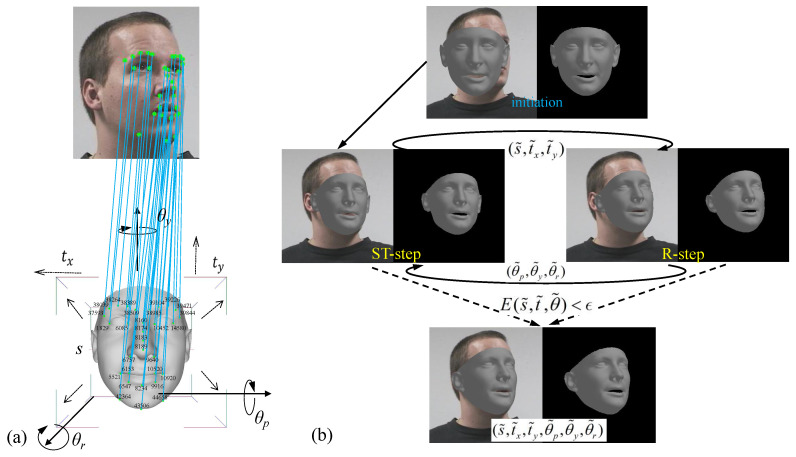
Overview of the proposed iterative optimization algorithm for 3D–2D keypoints matching. (**a**) The underlying head pose is inferred by 3D–2D keypoints matching. (**b**) A ST-step and a R-step are employed to alternatively optimize the 3D–2D keypoints matching.

**Figure 5 sensors-21-01841-f005:**
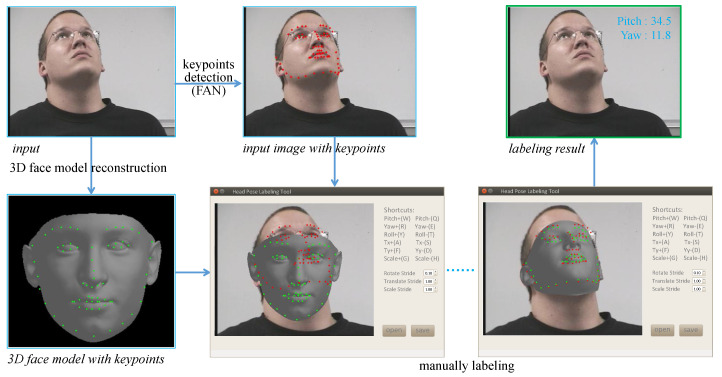
Pipeline of the ground-truth labels’ relabeling.

**Figure 6 sensors-21-01841-f006:**
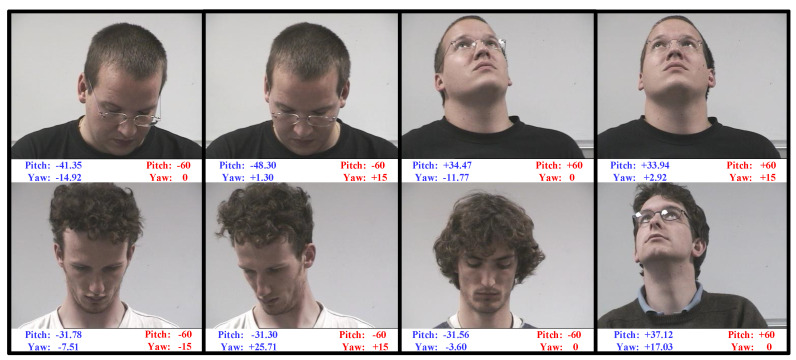
Original and relabeled head-pose angles of images randomly selected from Pointing’04. The original and relabeled angles are, respectively, marked in red and blue.

**Figure 7 sensors-21-01841-f007:**
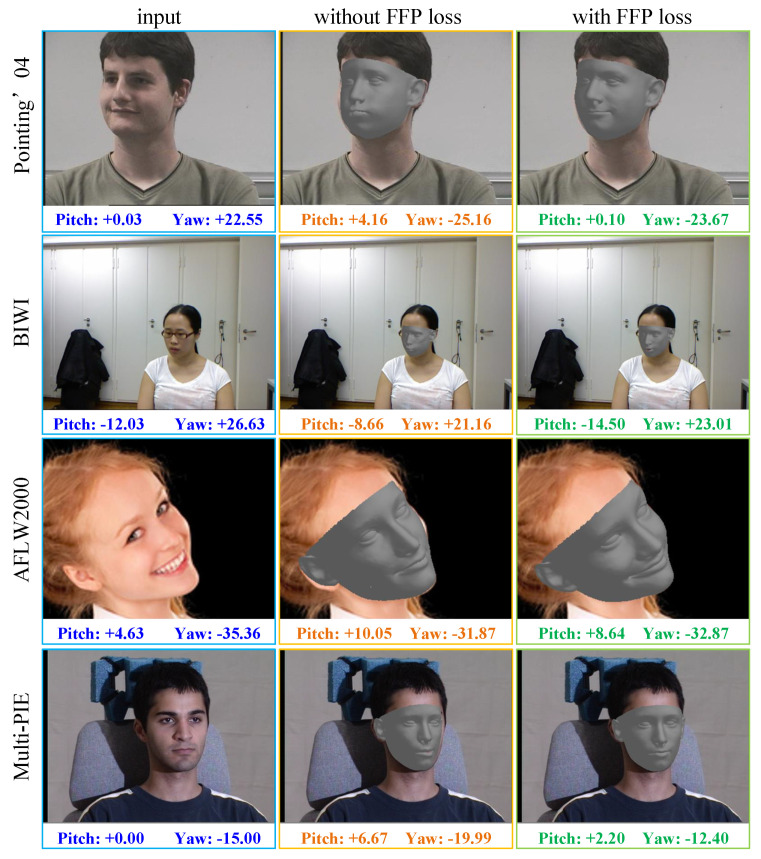
3D face models and estimated head-pose angles produced by the proposed method with/without FFP loss.

**Figure 8 sensors-21-01841-f008:**
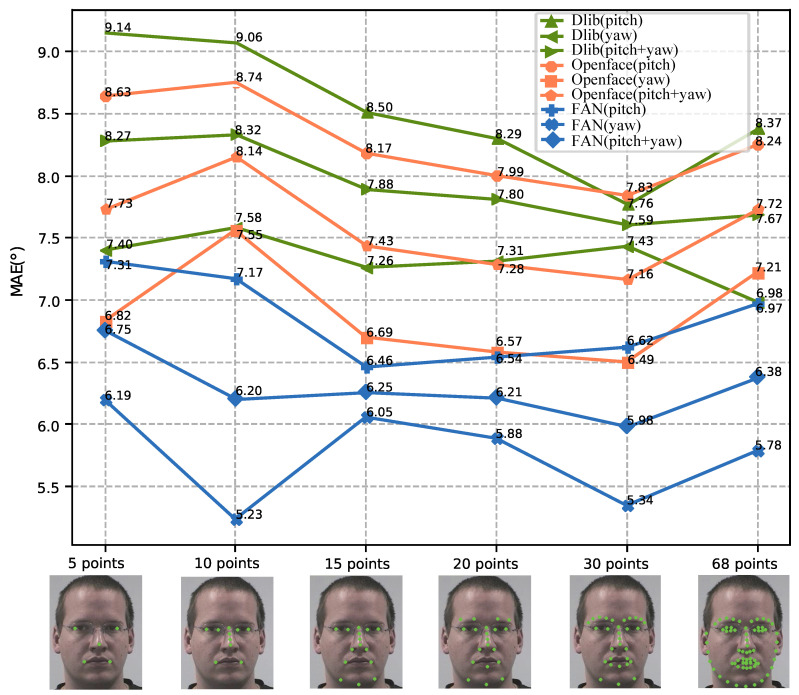
Performance of the proposed method with different facial keypoint detection methods under different keypoint configurations.

**Figure 9 sensors-21-01841-f009:**
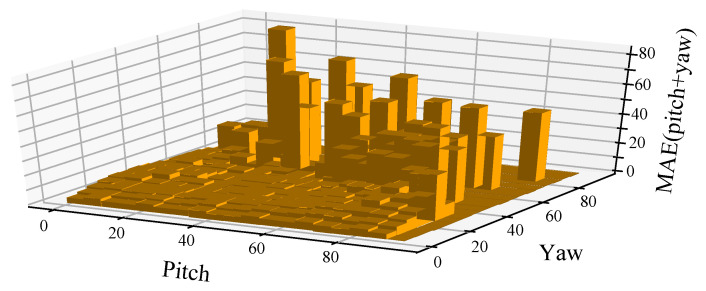
MAE distribution of different head pose angles.

**Figure 10 sensors-21-01841-f010:**
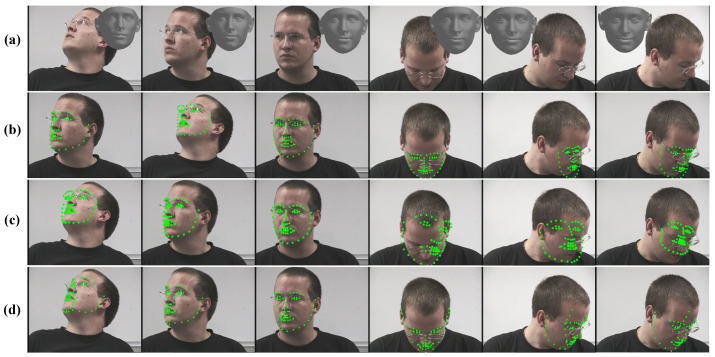
Reconstructed 3D face models and facial keypoints detected by three detectors under various head-pose angles. (**a**) The original images and the corresponding reconstructed 3D face models. (**b**) Keypoints detected by Dlib. (**c**) Keypoints detected by Openface. (**d**) Keypoints detected by FAN.

**Figure 11 sensors-21-01841-f011:**
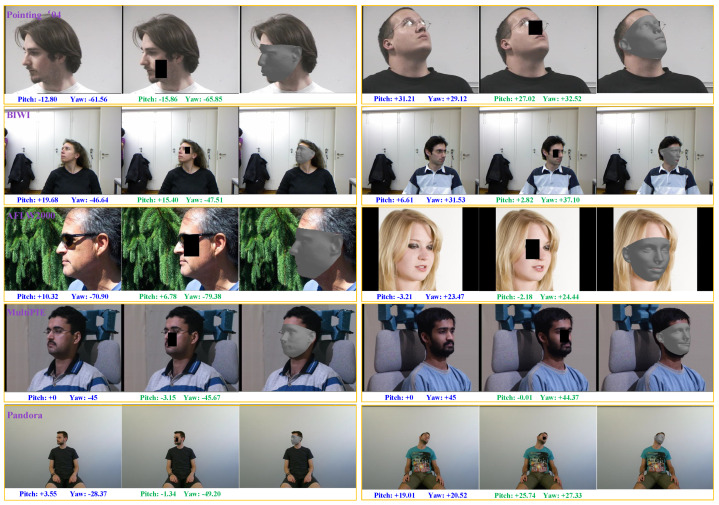
Example results produced by the proposed method on the occluded testing images. The ground-truth labels and the estimations are illustrated in blue and green, respectively.

**Figure 12 sensors-21-01841-f012:**
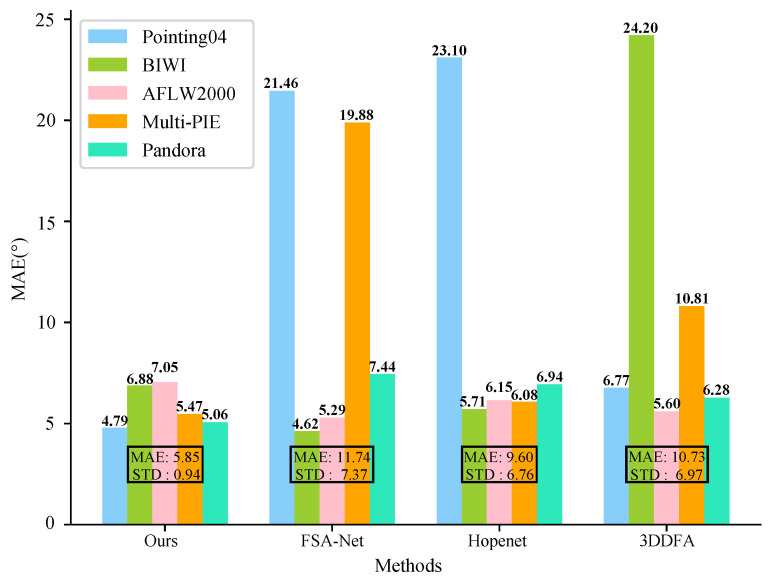
Cross-dataset experimental results. The average MAEs achieved by each method on each dataset are marked on the top of each result bar, and the MAEs and standard deviation reported by each method on all the datasets are marked across the result bars.

**Table 1 sensors-21-01841-t001:** MAEs of the proposed method with/without FFP loss for 3D face reconstruction.

Testing Dataset	Without FFP Loss			With FFP Loss		
	Pitch	Yaw	Pitch + Yaw	Pitch	Yaw	Pitch + Yaw
Pointing’04 [18]	7.42	6.59	7.01	5.27	4.30	4.78
BIWI [19]	9.65	6.79	8.22	7.94	5.81	6.88
AFLW2000 [20]	10.46	6.86	8.66	8.49	5.60	7.05
Multi-PIE [21]	7.12	7.87	7.50	5.77	5.17	5.47

**Table 2 sensors-21-01841-t002:** MAEs of the proposed method on testing images with/without occluded block (OB).

Testing Dataset	MAE(${}^{\circ }$) without OB			MAE(${}^{\circ }$) with OB		
	Pitch	Yaw	Avg	Pitch	Yaw	Avg
Pointing’04 [18]	5.27	4.30	4.78	6.32	5.85	6.09
BIWI [19]	7.94	5.81	6.88	11.91	5.88	8.90
AFLW2000 [20]	8.49	5.60	7.05	12.17	7.91	9.04
Multi-PIE [21]	5.77	5.17	5.47	5.12	9.83	7.48
Pandora [22]	4.99	6.33	5.66	6.98	6.60	6.79

**Table 3 sensors-21-01841-t003:** Computational load of the modules in our method.

Module	Frames per Second
3D face reconstruction(3DFR)	0.55
2D keypoints detection	14.49
3D-2D keypoints matching	46.31
Whole system	0.52
Whole sysem(wo 3DFR)	11.11

**Table 4 sensors-21-01841-t004:** Performance comparison of different head-pose estimation methods on the Pointing’04 dataset.

	MAE(${}^{\circ }$)		
	Yaw	Pitch	Avg
MGD [53]	6.90	8.00	7.46
kCovGa [54]	6.34	7.14	6.74
CovGA [54]	7.27	8.69	7.98
CNN [55]	5.17	5.36	5.27
fuzzy [56]	6.98	6.04	6.51
MSHF [57]	-	-	6.60
SAE-XGB [58]	6.16	7.17	6.67
hGLLiM [59]	7.93	8.47	8.20
Hopenet [26]	26.61	19.59	23.10
FSA-Net(FAN) [27]	25.90	18.01	21.96
3DDFA [37]	6.18	7.38	6.77
4C_4S_var4 [36]	10.63	6.34	8.49
ours(FAN-30)	4.30	5.27	4.78

**Table 5 sensors-21-01841-t005:** Performance comparison of different head-pose estimation methods on the BIWI dataset.

	MAE(${}^{\circ }$)			
	Yaw	Pitch	Roll	Avg
CNN-syn [30]	11.35	9.65	10.42	10.47
DNN [60]	11.89	7.12	12.78	10.60
regression [61]	8.85	8.70	-	8.78
Two-Stage [62]	9.49	11.34	6.00	10.41
KEPLER [25]	8.08	17.28	16.20	13.05
QuatNet [63]	4.01	5.49	2.94	4.15
Dlib [51]	16.76	13.80	6.19	12.25
FAN [48]	8.53	7.48	7.63	7.88
3DDFA [37]	36.18	12.25	8.78	19.07
QT_PYR [35]	5.41	12.80	6.33	8.18
QT_PY+R [35]	6.28	14.95	4.12	8.45
4C_4S_var4 [36]	6.21	3.95	4.16	4.77
Haar-Like(LBP) [64]	9.70	11.30	7.0	9.33
HAFA [65]	8.95	6.80	-	7.88
Ours(FAN-30)	5.81	7.94	6.74	6.83

**Table 6 sensors-21-01841-t006:** Performance comparison of different head-pose estimation methods on the Pandora dataset.

Methods	Input	Cropping	Fusion	MAE (${}^{\circ }$)			
				Pitch	Roll	Yaw	Avg
SingleCNN [22]	depth	X	-	8.1	6.2	11.7	8.67
	depth	✔	-	6.5	5.4	10.4	7.43
	FfD	✔	-	6.8	5.7	10.5	7.67
	gray-level	✔	-	7.1	5.6	9.0	7.23
	MI	✔	-	7.7	5.3	10.0	7.67
DoubleCNN [22]	depth + FfD	✔	concat	5.6	4.9	9.8	6.77
	depth + MI	✔	concat	6.0	4.5	9.2	6.57
POSEidon [22]	depth + FfD + MI	✔	concat	6.3	5.0	10.6	7.30
	depth + FfD + MI	✔	mul + concat	5.6	4.9	9.1	6.53
	depth + FfD + MI	✔	conv + concat	5.7	4.9	9.0	6.53
Hopenet [26]	single RGB	X	-	5.62	6.69	8.49	6.94
FSA-Net [27]	single RGB	X	-	6.93	5.06	10.32	7.44
3DDFA [37]	single RGB	X	-	6.62	4.65	7.58	6.28
Ours(FAN-20)	single RGB	X	-	4.99	3.87	6.33	5.06
Ours(FAN-30)	single RGB	X	-	5.21	3.88	6.27	5.12
Ours(FAN-68)	single RGB	X	-	5.83	3.97	6.80	5.53

FfD:Face-from-Depth (gray-level image reconstructed from depth map), MI:Motion Images.

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
