# Peer review of "Head Pose Estimation through Keypoints Matching between Reconstructed 3D Face Model and 2D Image"

_sensors, 2021, doi:10.3390/s21051841_

Round 1

Reviewer 1 Report

The paper "Head Pose Estimation through Keypoints Matching between
Reconstructed 3D Face Model and 2D Image" presents a method to predict the 3D head pose relying on RGB images. Specifically, a 3D model of the detected face is generated and then facial landmarks detected on the RGB image are matched with the ones of the model. In this way, authors note that the proposed method is not based on the accuracy of ground-truth labels of the training data.

The paper is well organized and experiments are well reported and discussed.
In general, the paper seems to be relevant for this research field.

However, in my opinion there are some issues that authors can take into account for the final version of the paper:

  1. Related Work section can be extended, including also the analysis of methods based on depth and 3D data, since the proposed framework exploits 3D face models. In particular, I think that these papers can be included:
    - Robust model-based 3d head pose estimation. (ICCV 2015)
    - Real-time 3d head pose and facial landmark estimation from depth images using triangular surface patch features (CVPR 2015)
    - POSEidon: Face-from-Depth for Driver Pose Estimation (CVPR 2017)
    - Real time head model creation and head pose estimation on consumer depth cameras (3DV 2014)
    Maybe these works can be included also in the table with the result comparisons.
  2. Why are two different networks (ResNet-18 and ResNet-101) used in order to extract the geometric and the expression parameters? Is the training procedure the same? What are the results obtained exploiting only the same models for both types of parameters?
  3. I believe that a great improvement of the paper can be the inclusion of tests about the performance of the proposed system in presence of face occlusions. In that context, the Pandora dataset (Face-from-depth for head pose estimation on depth images,  TPAMI 2018) can be used to test the system. What are the performance of facial landmark detectors in presence of face occlusions?
  4. What is the computational load of the presented framework? What is the performance in terms of frames per seconds? Can the system be used in real-time and real-world applications?

Reviewer 2 Report

This paper presents a method to estimate the head pose based on the matching between keypoints obtained in the 2D input image and a 3D face model.

The paper is well written, clearly structured following a traditional approach. Title, Abtract, Introduction and Related work sounds good and cover the most important details about the topic of the manuscript.

The methodology sounds correct and well justified. I suggest the release of the implementation as a valuable contribution to the science.

The authors present and extensive set of experiments to validate the proposed method, and the problem of inaccurate labels in head pose estimation datasets was successfully tackled.

Conclusions are supported during the document and references seems complete and relevant. I found only this relevant recent work: https://www.researchgate.net/publication/340659189_Head_posture_detection_with_embedded_attention_model

There are minor English typos, and it is not possible to read the whole caption of Figure 2. The size of the images and the width of the text do not match, but I understand this can be easily correct if the document will be accepted.

Reviewer 3 Report

The paper propose head pose estimation method based on key points matching between a face image and a 3D model of the face reconstructed using this face image. 

The paper considers the problem of estimation of a head orientation having one image of the head. And the improved accuracy of a head pose estimation is presented as a advantage of the proposed technique.

Meanwhile saying about object (head) orientation, the authors do not give the determinations of the head pose and system coordinates used for measuring estimated angles. The only Fig. 4a is not enough for clear explanation of the proposed method. Also from introduction it is not clear for what applications accurate head pose estimating is so important to accurate estimate and why. 

The Introduction provides representative overview of the relevant papers, but it misses at least two state-of the-art papers that consider similar problems. These papers are:

Gecer, Baris and Ploumpis, Stylianos and Kotsia, Irene and Zafeiriou, Stefanos, GANFIT: Generative Adversarial Network Fitting for High Fidelity 3D Face Reconstruction, Proceedings of the IEEE/CVF Conference on Computer Vision and Pattern Recognition (CVPR), 2019

Feng Y., Wu F., Shao X., Wang Y., Zhou X. (2018) Joint 3D Face Reconstruction and Dense Alignment with Position Map Regression Network. In: Ferrari V., Hebert M., Sminchisescu C., Weiss Y. (eds) Computer Vision – ECCV 2018. ECCV 2018. Lecture Notes in Computer Science, vol 11218. Springer, Cham. https://doi.org/10.1007/978-3-030-01264-9_33

It is worth to reflect why the proposed approach is better than existing frameworks.

The paper needs clarifying in the following points:

1) How 3D key points are detected in the reconstructed 3D model.

2) How 2D key points are detected in the input image.

3) Head orientation is described by three angles: pitch, yaw and roll. Why all results are given only for pitch and yaw.

4) Why MAE is chosen as performance metric.

5) How the author compared results in cases of data given in discrete form (as in the Pointing’04 dataset).

6) What contributions of the study allow to improve the performance.

7) In the discussion and conclusion it is worth reflecting what are advantages of the proposed method, not just repeating the main contributions and results.

Minor remarks:

line 7-8: The phrase is not complete.

Fig. 2: Text is out the the page borders.

line 193: MSE needs to be introduced before using the abbreviation.

Fig. 4: What is "the T.s-step"?

Round 2

Reviewer 1 Report

Authors have answered my questions and I am satisfied of the update version of the paper.

Author Response

Thank you!

Reviewer 3 Report

Dear Authors,

Thank you for your efforts in improving the paper. 

But unfortunately I did not find clear answers to my main questions and concerns. 

Thank you for the illustration Fig. R1 taken from your reference [16] (Erik, M.C.; Trivedi, M.M. Head Pose Estimation in Computer Vision: A Survey. TPAMI 2009, 31, 607–626). But it does not clarify in what system of coordinates you estimate the pose of a head: what is the origin of coordinates, how axes are directed, what is the sequence of angles' rotations.

It is hard to evaluate how good are the technique and the results of head pose estimation without clear determination of what is estimated.

Also it is still unclear why it is important for target application to estimate head pose with MAE, say, 4.79 degrees comparing with existing MAE of 6.5 degrees. 

Now the main steps of the proposed pipeline look like as the following:

1) For a given input image key points are found by CNN-based facial key points detector FAN.

2) Head 3D model is generated for given input image using BaselFace model and FaceWarehouse model. This step provides face 3D model and a set of key points for this model.

3) Then the problem of image orientation basing on a set of corresponding points in the 3D model and in the image. This step is performed iteratively searching the minimum of energy function (12). It provides head pose parameters.

For each step the authors proposed some modifications of the baselines, but  the overall technique seems has a lack of novelty.

Also some discussion is awaited on what contribution provides the advantages in performance of the proposed technique. 
